# Ethiopia National Food and Nutrition Survey to inform the Ethiopian National Food and Nutrition Strategy: a study protocol

Meseret Woldeyohannes,[1] Meron Girma  ,[1] Alemnesh Petros,[1] Alemayehu Hussen,[1] Aregash Samuel,[1] Danial Abera Dinssa,[1] Feyissa Challa,[1] Arnaud Laillou,[2] Stanley Chitekwe,[3] Kaleab Baye,[4] Ramadhani Noor,[3] Anne Sophie Donze,[3] Getachew Tollera,[1] Mesay Hailu Dangiso,[1] Lia Tadesse,[5] Meseret Zelalem,[5] Masresha Tessema  [1]

¹Food Science and Nutrition Research Directorate, Ethiopian Public Health Institute, Addis Ababa, Addis Ababa, Ethiopia
²UNICEF, Dakar, Senegal, Dakar, Senegal
³UNICEF Ethiopia, Addis Ababa, Ethiopia
⁴Addis Ababa University, Addis Ababa, Ethiopia
⁵Ethiopia Ministry of Health, Addis Ababa, Ethiopia

**Correspondence to**
Dr Masresha Tessema;
dr.masresha.tessema@gmail.com

## ABSTRACT

**Introduction** Ethiopia has made significant progress in reducing malnutrition in the past two decades. Despite such improvements, a substantial segment of the country's population remains chronically undernourished and suffers from micronutrient deficiencies and from increasing diet-related non-communicable diseases such as diabetes, hypertension and cancer. This survey aims to assess anthropometric status, dietary intake and micronutrient status of Ethiopian children, women and adolescent girls. The study will also assess coverage of direct and indirect nutrition-related interventions and map agricultural soil nutrients. The survey will serve as a baseline for the recently developed Ethiopian Food System Transformation Plan and will inform the implementation of the National Food and Nutrition Strategy.

**Methods and analysis** As a population-based, cross-sectional survey, the study will collect data from the 10 regions and 2 city administrations of Ethiopia. The study population will be women of reproductive age, children aged 0–59 months, school-aged children and adolescent girls. A total of 16 596 households will be surveyed, allowing the generation of national and regional estimates. A two-stage stratified cluster sampling procedure will be used to select households. In the first stage, 639 enumeration areas (EAs) will be selected using probability-proportional-to-size allocation. In the second stage, 26 eligible households will be selected within each EA using systematic random selection. Primary outcomes include coverage of direct and indirect nutrition interventions, infant and young child feeding (IYCF) practices, food insecurity, dietary intakes, mental health, anthropometric status, micronutrient status and soil nutrient status.

**Ethics and dissemination** The protocol was fully reviewed and approved by the Institutional Review Board of the Ethiopian Public Health Institute (protocol no: EPHI-IRB-317–2020). The study is based on voluntary participation and written informed consent is required from study participants. The findings will be disseminated via forums and conferences and will be submitted for publication in peer-reviewed journals.

---

### STRENGTHS AND LIMITATIONS OF THIS STUDY

⇒ The survey covers a large geographical area, collecting data on anthropometric status, 24 hours recall quantitative dietary intakes and the determination of micronutrient status in the same participants or household, while also capturing data on the food system in Ethiopia.

⇒ The study aims to improve understanding of nutritional problems across multiple facets—from agricultural soil to people to the environment in Ethiopia.

⇒ Inherent to the cross-sectional design of the study, the findings of this study cannot be used to establish cause and effect.

⇒ The study design prevents us from considering seasonal differences in nutritional outcomes and determinants.

---

## INTRODUCTION

Globally, one in every three persons is affected by one of more forms of malnutrition.[1] Women and children are particularly vulnerable to malnutrition due to increased physiological nutrient needs required to support fetal and child growth.[2] Nutritional deprivation during early life impairs growth and development, leading to poor school performance, reduced productivity and loss of earnings in later life.[3] Consequently, the first 1000 days of life, from conception to the child's second year of life, were recognised as a critical window of opportunity to effectively prevent malnutrition.[3 4] Adolescence is also identified as a second window of opportunity to correct nutritional inadequacies and adversities faced in early life, but little is known about this life stage.

Despite significant progress over the past two decades, the burden of malnutrition in Ethiopia remains high.[5–7] Nationally, 37%

of Ethiopian children under 5 years of age are stunted,[7] and 22% of women of reproductive age (WRA) are chronically undernourished (body mass index (BMI) <18.5 kg/m²).[5] Only 14% of children under 2 years of age consumed the minimum number of recommended food groups.[5] Furthermore, micronutrient deficiencies coexist with chronic energy deficiency.[8] This along with the ongoing nutrition transition, characterised by shifts in diets,[9] is further complicating the nutrition landscape by increasing the prevalence of overweight and non-communicable diseases.[5] Nearly a fifth (16%) of Ethiopian adults are estimated to be hypertensive, and 3% are diabetic.[10] Therefore, addressing not only undernutrition but all forms of malnutrition is critical.

The Sustainable Development Goals (SDGs) recognise the importance of nutrition, primarily driven by the need to mitigate its detrimental consequences. Further, the 2012 World Health Assembly identified global targets to be achieved by 2025 that aim to reduce stunting, anaemia, low birth weight and childhood obesity. These targets are used to track progress in SDG 2: zero hunger.[11] Recognising the importance of good nutrition, the Government of Ethiopia has made ending malnutrition a national priority. Ethiopia started implementing its first National Nutrition Program in 2008.[12] The second phase of this programme (2011–2016) was a multisectoral programme aimed at accelerating progress in reducing malnutrition.[13] Moreover, Ethiopia's first Food and Nutrition Policy was endorsed in 2018,[14] followed the National Food and Nutrition Strategy[15] which was launched in 2021 to provide a framework for the operationalisation of the policy. Acceleration of progress in the reduction of malnutrition requires the design and implementation of direct and indirect nutrition interventions that can be implemented at scale. To this end, understanding the various factors contributing to the different forms of malnutrition is critical.

Multiple factors operating at the immediate, underlying and basic levels contribute to malnutrition.[2] Inadequate dietary intake and poor health are immediate determinants.[2] Household food security, child care practices, access to health services and healthy environments are underlying determinants.[16] Structural and contextual factors such as economic structures, and political, environmental, social and cultural factors are the basic determinants of malnutrition.[2] The contribution of these factors varies across different contexts, and target groups, but studies capturing all these factors in a single survey are scant. The lack of timely and comprehensive information on nutritional status across critical life stages and their determinants is a bottleneck that is preventing Ethiopia from designing effective interventions. Up-to-date and comprehensive data on the coverage of direct and indirect nutrition interventions delivered across various implementing sectors of the National Food and Nutrition Strategy are not yet available. This is unfortunate as such data could inform the implementation of the strategy, but it can also serve as a baseline against which progress can be tracked.

Therefore, this study aims to provide the first ever comprehensive information on the nutritional status of different populations in Ethiopia to support evidence-based implementation of the National Food and Nutrition Strategy.

## Objectives
The overall goal of this study will be to produce nationally and regionally representative estimates on anthropometric status, coverage of nutrition interventions, dietary intakes, and micronutrient status for children, adolescent girls and WRA in Ethiopia.

Specific objectives include:
1. Assess the coverage of direct and indirect nutrition interventions.
2. Assess food consumption patterns and nutrient intake of children aged 6–59 months and WRA.
3. Assess the micronutrient status of children (vitamin A, anaemia, iron, iodine and zinc), adolescent girls and WRA (vitamin A, vitamin D, anaemia, iron, iodine, zinc, folate, vitamin $B_{12}$)
4. Assess the anthropometric status of children under 5 years of age, school-age children (6–12 years), adolescent girls and WRA.
5. Assess the geographical distribution of soil micronutrient status in the Ethiopian agricultural soil.

## METHODS AND ANALYSIS
### Study design
This study is a nationally and subnationally (regionally) representative cross-sectional survey that will characterise dietary intake, micronutrient status and access to nutrition-related services for different target populations. Given that soil nutrient content can influence micronutrient content of foods and hence affect nutrient intake, the soil nutrient composition will also be analysed. The study will have four main components. The first component will assess nutrition-specific and nutrition-sensitive indicators (NSS) for all target groups (children aged 0–59 months and WRA, school-age children and adolescent girls) using semistructured questionnaires. The second component will measure quantitative dietary intake for children aged 6–59 months and WRA (15–49 years). The third component of the survey will collect biomarker samples from all children (6–59 months), school-age children (6–12 years), adolescent girls (10–19 years) and WRA (15–49 years). The final component of the study will measure micronutrients in agricultural soils. The study data will be collected from July 2021 to December 2023.

### Setting
Ethiopia has an estimated population size of 120 million and is the second most populous country in Africa.[17] The majority of its population resides in rural areas (70%).[17] Agriculture accounts for 40% of the country's gross domestic product.[17] Children aged 15 years and younger make up 40% of the Ethiopian population in 2021.[18] Ethiopia is administratively divided into 10 regions and 2 city

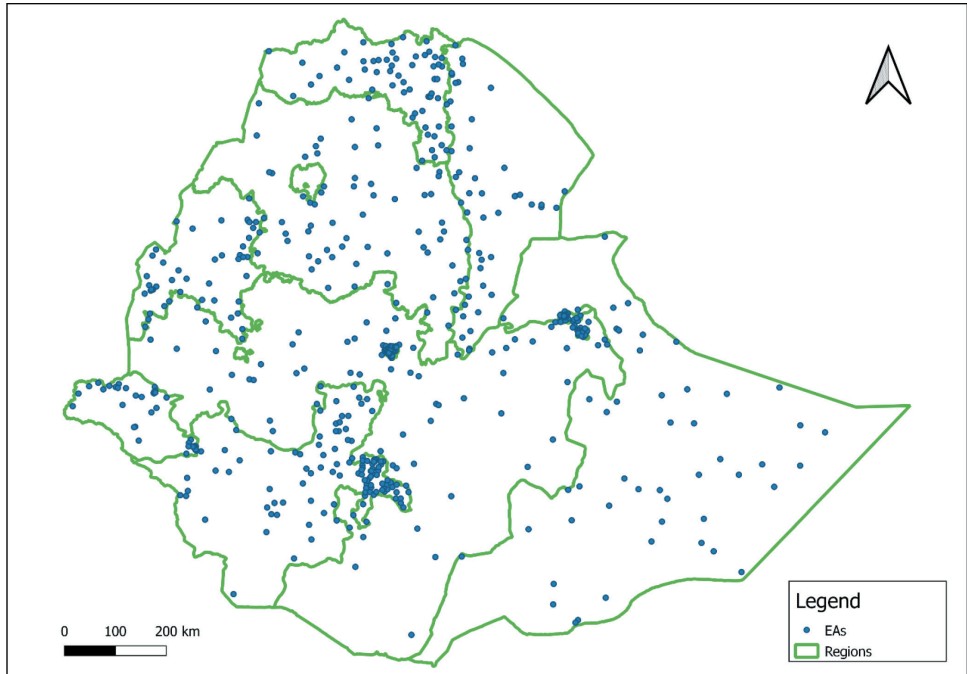

**Figure 1** Map showing study enumeration areas (EAs) across regions.

administrations. This study will be conducted in all the regions and city administrations of the country. Figure 1 provides a geographical representation of the study areas.

### Participants

The target population of this study are (1) WRA aged 15–49 years, (2) Children aged 0–59 months, (3) School-age children aged 6–12 years, (4) Adolescent girls aged 10–19 years and (5) Household heads.

### Sample size calculations

Sample size was estimated to guarantee adequate precision to generate national and regional estimates for selected indicators for each study target group. Indicators used for each target group are shown in online supplemental table S1. The required number of households and target groups was calculated using a single population proportion formula at the regional level. We used region-specific prevalence estimates for indicators, a 5% margin-of-error, a design effect of 1.5, a household response rate of 95% and an individual response rate of 80%. The initial sample size was then adjusted for region-specific average household size and percentage of the target population from the total population. An indicator that provides the maximum number of households was used to estimate the final sample size for each region. Regional sample sizes were summed up to derive the total (national) sample size. Based on these calculations, the total sample size for the overall survey was 16 596 households (online supplemental table S2).

For WRA, dietary and biomarker data will be collected in half of the selected households within each enumeration area (EA). This selection will yield a total sample size of 7386 WRA (50% of the expected 14 772 WRA). The sample size needed to assess dietary intakes and micronutrient status of WRA was calculated using the prevalence of inadequate zinc intake, which yielded the largest sample size.[8 19]

### Sampling procedures

A two-stage stratified cluster sampling procedure will be used to select households. In the first stage, 639 EAs, 257 urban and 382 rural, will be selected using probability-proportional-to-size allocation. We will use the 2018 Ethiopia Population and Housing Census EAs sampling frame to select EAs (the primary sampling units). The Central Statistical Agency prepared the EAs sampling frame. An EA typically contains 100–150 households. EA maps will be used to delineate the boundaries of the selected EA. In the second stage of sampling to identify eligible households, all households with the EA will be listed. A household will be eligible for selection if at least one of the study target groups are residents (de jure) or stayed at the household the night before the interview (de factor).

Twenty-six (26) eligible households will be selected within each EA using systematic random selection. All target groups will be eligible for the NSS interview in the selected households. All children aged 6–59 months will also be eligible for dietary assessment. Women residing in 13 households (out of 26 households) who will be selected randomly will be eligible for dietary assessment. Biomarker samples will be collected for all children under 5 years of age, school-age children and adolescents in the selected households. Similar to dietary assessment, biomarker samples will be collected for women residing in half of the selected households (figure 2).

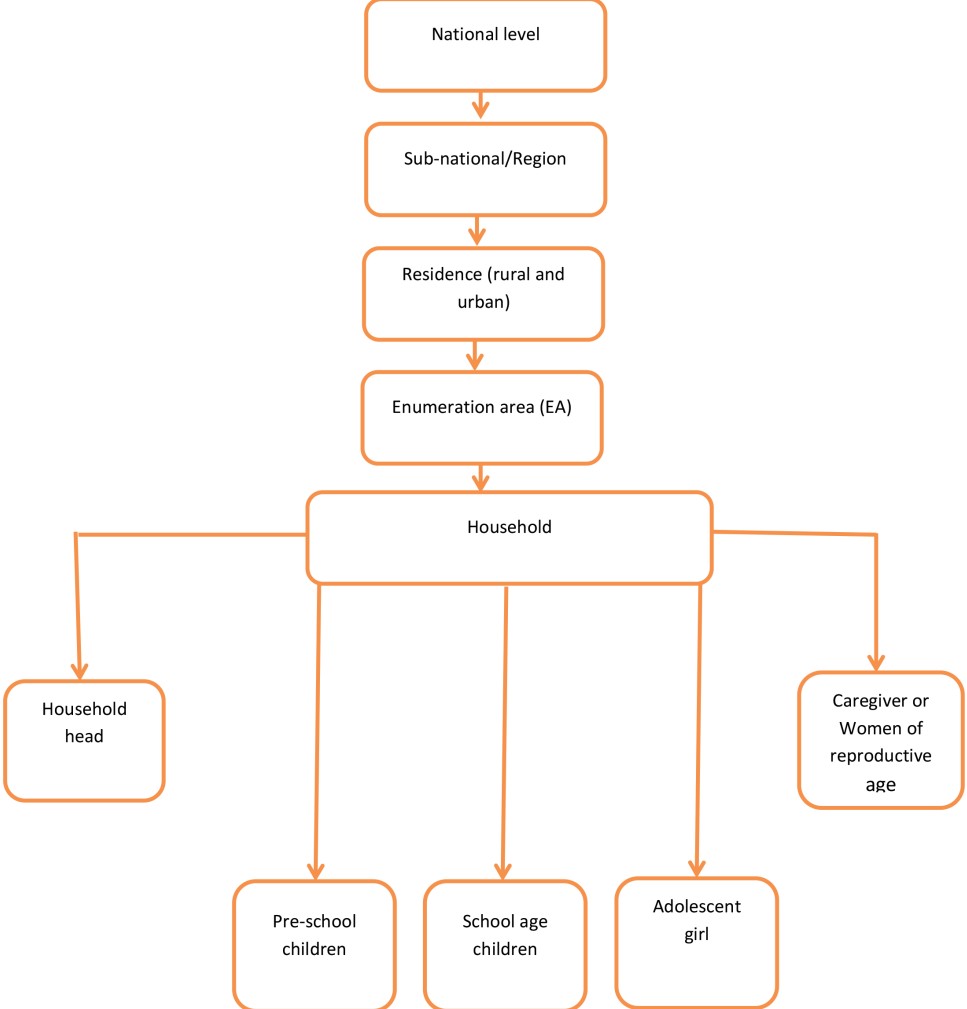

**Figure 2** Sampling frame for the Ethiopia National Food and Nutrition Survey to inform the Ethiopian National Food and Nutrition Strategy.

## Outcomes
### Coverage of direct and indirect nutrition interventions
A structured questionnaire will be used to determine the coverage of direct and indirect nutrition interventions provided to children aged 6–59 months, WRA and adolescent girls. Direct nutrition interventions included vitamin A supplementation, iron supplementation, zinc supplementation, growth monitoring and promotion, nutrition counselling services, and food fortification. Water, sanitation and hygiene, coverage of food or cash assistance programme, women empowerment, and mental health will be some nutrition-sensitive indicators considered in this study (table 1). We will use standard indicator definitions proposed by the Data for Decisions to Expand Nutrition Transformation project (DataDENT) to assess coverage of nutrition programmes.

### Anthropometric status
Using standardised procedures, anthropometric measurements, including weight, height/length and mid-upper arm circumference, will be taken for all study target populations.[20] Anthropometric indices (weight-for-height z-scores, length/height-for-age z-scores, weight-for-age z-scores, BMI-for-age z-scores) will be calculated using the WHO 2006 child growth standards and the WHO 2007 child growth reference data. Stunting (length/height-for-age z-scores below −2 SD), wasting (weight-for-height z-scores below −2 SD) and Mid-Upper Arm Circumference (MUAC), underweight (weight-for-age z-scores below −2 SD), thinness (BMI-for-age z-scores below −2 SD) and BMI will be the primary anthropometric outcomes of interest.

### IYCF practices
IYCF practices will be assessed using the new WHO and UNICEF recommended 17 indicators to evaluate IYCF practices.[21]

### Food insecurity
The Food Insecurity Experience Scale (FIES) will be used to assess household food security.[22] The FIES consists of eight questions that assess household experience related to adequate food access. Experience questions range from worrying about getting enough food to not eating for a whole day.

In addition to these outcome indicators, information on the sociodemographic characteristic of households,

**Table 1** Nutrition direct and indirect interventions coverage

| No | Indicator | Target population |
|---|---|---|
| | **Nutrition indirect intervention coverage** | |
| | *Child interventions* | |
| 1. | Children received iron tablets/syrup in the last 12 months | Children aged 6–59 months |
| 2. | Children received vitamin A supplements in the past 6 months | Children aged 6–59 months |
| 3. | Children received deworming tablets in the past 6 months | Children aged 24–59 months |
| 4. | All eight basic vaccinations: one dose of BCG, three doses of DPT, three doses of the polio vaccine, and one dose of the measles vaccine | Children aged 9–59 months |
| 5. | No vaccination | Children aged 0–59 months |
| | *Growth monitoring* | |
| 6. | Weight measured in the last 3 months | Children aged 0–23 months |
| 7. | Height measured in the last 3 months (optional) | Children aged 0–23 months |
| 8. | MUAC measured in the last 3 months (optional) | Children aged 0–23 months |
| | *Infant and young child feeding (IYCF) counselling* | |
| 9. | Mothers with children 6–23 months received any IYCF counselling | Children aged 6–23 months |
| 10. | Mothers with children 6–23 months received age-appropriate IYCF counselling | Children aged 6–23 months |
| | *Early breastfeeding counselling* | |
| 11. | Women *received breastfeeding counselling with observation during the first 2 days after birth* | Women aged 15–49 years with a live birth in the past 5 years for the most recent birth |
| 12. | Women *received breastfeeding counselling during the first month after birth* | Women aged 15–49 years with a live birth in the past 5 years for the most recent birth |
| | *Coverage of nutritional interventions during pregnancy/antenatal care (ANC)* | |
| 13. | Percentage of women who had four or more ANC visits for the most recent birth | Women aged 15–49 years with a birth in the last 5 years |
| 14. | Percentage of women who received counselling about healthy eating during pregnancy | Women aged 15–49 years who received ANC for their most recent birth |
| 15. | Percentage of women whose weight gain was monitored during pregnancy | Women aged 15–49 years who received ANC for their most recent birth |
| 16. | Women received food or cash assistance during pregnancy | Women aged 15–49 years with a birth in the last 5 years |
| 17. | Women took 90+ iron/folate tablets during pregnancy | Women aged 15–49 years with a live birth in the past 5 years for the most recent birth |
| 18. | Women received deworming tablets during pregnancy | Women aged 15–49 years with a live birth in the past 5 years for the most recent birth |
| | **Nutrition indirect intervention coverage** | |
| 19. | Basic water services | Household |
| 20. | Basic hygiene services | Household |
| 21. | Basic sanitation services | Household |
| 22. | Food insecurity (not a service hence no coverage) | Household |
| 23. | Women received food or cash assistance during pregnancy | Women aged 15–49 years with a live birth in the past 5 years for the most recent birth |
| 24. | Basic water services | Household |
| 25. | Basic hygiene services | Household |
| 26. | Basic sanitation services | Household |
| 27. | Presence of common mental health disorders in the past month | Women aged 15–49 years |
| 28. | Women empowerment | Women aged 15–49 years |
| 29. | Livestock ownership | Household |
| 30 | Agricultural productivity by food group | Household |

child health, maternal health, employment status and household agricultural practices will be collected using structured questionnaires.

## Mental health of women

Common mental health disorders will be assessed using the WHO Self-reporting Questionnaire which consists of 20 questions. Women will be classified as having a common mental health disorder if the row score will be greater or equal to 6 out of 20.[23]

## Assessment of dietary intakes of children and WRA

We will measure dietary intake for children aged 6–59 months and WRA. A 1 day quantitative multiple-pass 24-hour recall will be conducted to assess dietary intakes. The interactive multiple-pass 24-hour recall interview consists of four steps designed to enhance memory.[24] All days of the week will be proportionately represented during the dietary survey to account for the day of the week effects on food intake. To account for the day-to-day variability of dietary intake within individuals, a second non-consecutive day 24-hour recall (repeat) will be collected (within 2–10 days of the first recall) on a randomly selected subsample of WRA and children. The number of repeats needed is determined by allocating for each region 50 repeats, which is then multiplied by a design effect of 1.5 and a 10% non-response rate. The number of repeats will be rounded up to 1244 recalls for each target group to ensure that the minimum number of repeats (n=83) needed from each region would be collected. Detailed non-standard recipe ingredient data will be collected for all mixed dishes that were prepared at home.

We will use 15 food groups to assess the dietary intakes of women (15–49 months) and children aged 24–59 months. These food groups were: (1) Cereals and their products, (2) Starchy roots and tubers, and their products, (3) Pulses and their products, (4) Vegetables and their products, (5) Fruits and their products, (6) Meat and poultry their products, (7) Eggs and their products, (8) Fish, shellfish and their products, (9) Milk and milk products, (10) Fats and oils, (11) Nuts and seeds, (12) Sugar and sweetened products, (13) Beverages, (14) Spices and condiments, and (15) Miscellaneous. For children aged 6–23 months, we will use the updated WHO, UNICEF food groups: (1) Breast milk, (2) Grains, roots and tubers, (3) Pulses, nuts and seeds, (4) Dairy products, (5) Flesh foods (meats, fish, poultry, organ meats), (6) Eggs, (7) Vitamin-A rich fruits and vegetables, and (8) Other fruits and vegetables. These food groups were adapted from the FAO/WHO Global Individual Food consumption data Tool food groups.[25]

## Dietary assessment presurvey work

We carried out presurvey work to aid dietary data collection following recommendations set by the Intake: Centre for Dietary Assessment.[26 27] An initial step will be to develop a food and ingredient list containing a comprehensive list of food items, mixed dishes and ingredients expected to be consumed by the study target groups. The food list will be generated using data from the first 2011 Ethiopian National Food Consumption Survey.[19] Other common foods consumed across the regions in Ethiopia will be derived from the 2016 Household Income and Expenditure Surveys,[28] the Ethiopian Food Composition Tables, and dietary intake data from other recent dietary assessment surveys conducted by the Ethiopian Public Health Institute (EPHI). Portion size estimation methods suitable for large-scale studies will be preselected for use in the survey following intake recommendations.[29] The selected methods will be direct measurement of actual foods consumed, standard unit: size and number, proxy measurement using play dough, water, rice, and maize flour, and finally using food price to estimate the amount of food consumed. Portion size estimation methods will be assigned for all foods included in the food list.

## Assessment of micronutrient status

Blood specimens will be collected from the study population to determine serum retinol, ferritin, soluble transferrin receptor (sTfR), zinc, folate, vitamin $B_{12}$, red blood cell (RBC) folate and 25-hydroxyvitamin D. Additionally, markers of inflammation, alpha(1)-acid glycoprotein (AGP), high-sensitivity C reactive protein (hsCRP) will also be measured. We will also analyse parasites from stool specimens. All laboratory analyses will be performed at the EPHI Clinical chemistry, and Food Science and Nutrition Laboratories. Both laboratories participate in an external quality assessment scheme and are accredited by the Ethiopian National Accreditation Office. Collection, storage and analytical procedures for blood, urine, stool and salt samples are described below. The details of each biomarker analysis are described in online supplemental materials 1–11.

### Blood sample collection and analysis

Venous blood samples (5–7 mL) will be collected using vacutainer tubes following standard operating procedures.[30] Trace mineral-free vacutainer tubes will be used to collect blood for trace metal analysis. After collection, blood samples will be allowed to clot for 30 min in cold boxes (<8°C). Samples will then be centrifuged at 3000 rpm (revolution per minute) for 10 min. The separated serum will be aliquoted and stored in −20°C portable freezers in the field. Samples will then be transported to EPHI and stored at −80°C until analysis. Haemoglobin will be measured in the field using Hemocue (Hb 301, Hemocue AB, Angelholm, Sweden).[31 31] If the haemoglobin values are below WHO cut-off point(11 g/dL), the phlebotomist will send whole blood samples to the EPHI laboratory to identify haemoglobinopathies using the electrophoresis method.[32] Malaria test will be conducted onsite using Bioline Malaria Ag P.f rapid diagnostic test kits (RDT) for *Plasmodium*.[33] Serum sTfR, AGP, hsCRP, folate, RBC folate, vitamin $B_{12}$ and ferritin will be measured using Cobas 6000 analyzer (Roche Diagnostics

GmbH, Mannheim, Germany). Serum retinol will be measured using the high-performance liquid chromatography method,[33] and serum zinc and selenium will be measured using a microwave plasma atomic emission spectrometers analyser.

### Stool and urine sample collection and analysis

Stool samples will be collected using stool cups and stored in 10% formalin to preserve the parasite until analysis.[34] A portion of each stool sample will be used to detect direct ova, larvae and cysts of intestine parasites using formal ether concentration technique.[35] Urine samples will be collected from WRA and school-age children using 60 mL urine cup containers. Samples will be stored at −20°C. Urinary iodine excretion will be assessed by Sandell Kolthoff reaction at EPHI Laboratory using Shimadzu 1800 UV-Vis spectroscopy.[36]

### Salt collection and analysis

Salt samples will be collected from households with WRA for whom dietary data will be collected. At least 25 grams (one coffee cup) of salt will be collected to determine iodine content using the iodometric titration method.[37]

### Assessment of nutrients in the soil

Soil samples will be collected from three households in each EA. Zigzag or cross-sampling method will be used to collect 10–20 subsamples (0–30 cm depth) constituting one composite sample. Subsamples will be collected at a separation distance of 5 m. After thoroughly mixing the composite samples, 1 kg soil sample will be transferred to polyethylene bags. The collected soil samples will be air-dried in wooden trays and disaggregated using a ceramic mortar and pestle (soil grinder) at the EPHI soil laboratory. Samples will then pass through a 6 mm sieve of stainless steel screens to remove debris and homogenise the soil sample. The sieved fraction will be further pulverised to pass through a 1 mm sieve for the micronutrient analysis. Soil zinc, iron, copper and manganese will be determined following standard procedures.[38] Micronutrient content will be determined using inductively coupled plasma-optical emission spectroscopy after extraction with diethylene triamine penta acetic acid. Additional variables that affect the mobility of micronutrients in the soil and their uptake into crops will also be measured. These variables include soil reaction (pH), electrical conductivity, organic matter, total nitrogen and soil organic carbon content. Data collectors will also record topography, slope, cropping history, type and fertiliser application information. Table 2 provides a summary of procedures for each of the four components of the survey by study target groups.

### Data quality assurance and analysis

Training of trainers on components of the survey will be held before training the data collectors and supervisors. After 15 days of training on methodological

| Table 2 | Summary of data collection procedures for each of the four components of the survey | | | | | | |
|---|---|---|---|---|---|---|---|
| | Child 0–5 months | Child 6–23 months | Child 24–59 months | School children 6–12 years | Adolescent girls 10–19 years | WRA 15–49 years | Household |
| **Nutrition direct and indirect intervention indicators** | | | | | | | |
| Infant and young child feeding practices | X | X | | | | | |
| Nutritional information for adolescent girls | | | | | X | | |
| Food insecurity | | | | | | | X |
| Water, sanitation and hygiene practices | | | | | | | X |
| Coverage of food fortification | | | | | | | X |
| Agricultural practices | | | | | | | X |
| Mental health | | | | | X | | |
| Anthropometric status | X | X | X | | X | X | |
| **Dietary assessment** | | | | | | | |
| 24 hours recall quantitative dietary intake | | X | X | | | X | |
| **Assessment of biomarker status** | | | | | | | |
| Blood sample | | X | X | X | X | X | |
| Urine sample | | | | X | | X | |
| Stool sample | | X | X | X | X | X | |
| Salt sample collection | | | | | | | X |
| **Assessment of micronutrients in the soil** | | | | | | | |
| Soil micronutrient assessment | | | | | | | X |

WRA, women of reproductive age.

procedures, questionnaires and quality assurance, the questionnaires will be tested in a pilot group (in EAs not included in the actual survey), and adapted based on the received feedback from the survey team. The questionnaires (including the food list) were translated into local languages (Amharic, Oromifa, Tigrigna, Somali and Afar) and back-translated to English to ensure the quality of the translation. The data collectors' measurements will be standardised to ensure that the interobserver variability is within tolerable limits. Supervisors received additional training on teamwork and on monitoring and supervising the data collection process. All data collection tools are programmed using open-source software (ODK) (online supplemental file 12). Data quality checks will be included during ODK programming to prevent data recording errors. These include restricted responses, filter insert choices, skip patterns and defaults. During data collection daily data tracking forms will be completed to track completed surveys for each study component to prevent missing data. High frequency checks will be identified prior to the surveys, and error tracking forms will be designed to track data quality in real time. These checks included completeness checks, target group tracking and duplicate ID checks. Random field supervision visits will also be made to check data quality. Every day, collected data will be sent to the EPHI central server and imported into statistical software programs as comma-separated values files. For laboratory analysis, a quality control chart will be used to ensure the internal and external quality control materials are in the acceptable range.

The primary data analysis will focus on computing frequencies and percentages for categorical variables and summary statistics (like means, medians SD, IQR) for summarising continuous variables. Sample weights will be constructed based on the selection probabilities of EAs, eligible households and non-response rates. All analyses will also be adjusted for the survey design. Additional subgroup analysis will be computed for variables with adequate sample sizes for each category. The Biomarkers Reflecting Inflammation and Nutrition Determinants of Anemia working group's regression correction approach will be used to account for inflammation in the study of all micronutrients status using the biomarkers CRP and AGP. Geostatistical analyses will be employed to determine the spatial patterns of micronutrient distribution in the soil and blood samples. The wealth index will be constructed using principal component analysis.[39] The Rasch model will be used to construct the FIES.[22] All analyses will be done using STATA V.16 and Aeronautical Reconnaissance Coverage Geographic Information System (ArcGIS). Anthropometric indices will be calculated using the WHO Anthro software for children under 5 years of gae and WHO AnthroPlus software for adolescents.

**Patient and public involvement**
None.

**Ethics and dissemination**
The study protocol is approved by the Institutional Review Board of the EPHI (protocol no: EPHI-IRB-317–2020). Written informed consent will be obtained from each respondent and participants may withdraw at any time (online supplemental material 13). Confidentiality of all collected data will be given high priority during each stage of data handling. Individual names and personal information of respondents will be kept confidential and data sets will be kept anonymous for analysis.

The study's findings will be disseminated through several communication channels, including stakeholder workshops, various local and international conferences, and technical reports. Additionally, the findings will be submitted for publication in peer-reviewed journals.

**DISCUSSION**
This comprehensive, nationally representative survey will for the first time characterise simultaneously the dietary intake and micronutrient status of Ethiopian children, adolescent girls and WRA. Besides, the study assesses key drivers of malnutrition including soil nutrient composition, as well as coverage of direct and indirect nutrition interventions. The survey will provide key insights informing the implementation of Ethiopia's National Food and Nutrition Strategy.

High-quality and timely data are critical to assess the burden of nutritional problems, identify vulnerable populations and priority actions, track the implementation of nutrition programmes, and assess impact.[40 41] Ethiopia conducted its first ever food consumption survey in 2011[19] and its micronutrient survey in 2015.[42] Both surveys were collected at different times, which made it difficult to link the two surveys. The causes of malnutrition are numerous and often interconnected, and addressing this problem requires a comprehensive and multisectoral approach. One of the key challenges in addressing malnutrition is the lack of data on multiple indicators from various sectors. This data is essential for understanding the underlying causes of malnutrition and developing effective strategies to address it. For example, data on soil nutrient levels is critical for understanding the nutritional quality of the crops that are grown, while data on diets and micronutrient status is essential for understanding the nutritional status of individuals and populations. Exposure to direct and indirect nutrition interventions is also important in addressing malnutrition. By collecting data on multiple indicators from various sectors, policymakers and program implementer can develop evidence-based strategies to address malnutrition and improve the health and wellbeing of populations. In this regard, this survey is uniquely positioned to integrate data from multiple domains to support evidence-based decision making for improved diets, nutrition and overall well-being.

This study will allow us to evaluate progress relative to previous food consumption and micronutrient surveys, but, more importantly, will serve as a baseline against

which future progress related to the implementation of the National Food and Nutrition Strategy will be evaluated. Furthermore, the current survey will also serve as a baseline for the Ethiopian Food System Transformation Plan by capturing the majority of indicators used for monitoring food systems-related progress, thus filling information gaps that could have impeded successful implementation of the National Food and Nutrition Strategy. By establishing 13 strategic objectives, the National Food and Nutrition Strategy is intended to be aligned with the strategic directions of the Food and Nutrition Policy. Each strategy direction includes initiatives, actions and key performance indicators, as well as leading and collaborating sectors. The key performance indicators should be evaluated to determine the progress of each implementing sector's achievement. The current survey will provide up-to-date national and subnational information on the current food and nutrition situation in Ethiopia for different target populations as well as provide a comprehensive list of indicators that are pertinent to the implementation of the policy.[40] In addition, this study will provide information on context-specific determinants for prioritising direct and indirect actions that can be implemented across sectors taking into account the specific needs of different target populations.

Additionally, effective multisectoral interventions that address the immediate and underlying determinants of malnutrition must be implemented in order to accelerate the reduction of malnutrition in all its forms.[40] These interventions need to address context-specific determinants to reduce malnutrition effectively.[40] The lack of timely and disaggregated information on the determinants of malnutrition is a bottleneck to preventing malnutrition, particularly among the most vulnerable target populations. This study will also provide information on the coverage and quality of interventions which can be used to contextualise National Food and Nutrition Strategy monitoring frameworks, monitor implementation and track progress towards global and local targets.

Although this study will provide regionally and nationally representative estimates for key indicators and critical life stages, it has several limitations. Inherent to the cross-sectional design of the study, the findings of this study cannot be used to establish cause and effect. Additionally, the design prevents us from considering seasonal differences in nutritional outcomes and determinants. This study also relies on self-reported data, which are subject to recall bias. Notwithstanding the abovementioned limitations, this study is uniquely designed to combine the assessment of anthropometric status, 24-hour recall quantitative dietary intakes and the determination of micronutrient status in the same participants, while at the same time capturing data on the food system. Additionally, the study will be evaluating micronutrients in agricultural soil, which will expand our understanding of factors that influence nutrition. To the best of our knowledge, this will be among—if not—the first study to simultaneously collect these variables from the same household. This

could contribute to a better understanding of nutritional problems across multiple facets—from soil to people to the environment. In the past, nutrition programmes implemented in Ethiopia have relied on information provided from small-scale studies and population-based surveys such as the Ethiopia Demographic and Health Surveys.[5–7 43 44] Although these data sources provide some information to track progress and tailor interventions, they only provide data on a limited number of nutrition indicators and do not measure dietary intakes and assess biomarker status. This study will fill these data gaps by providing information on comprehensive indicators that show the burden and spatial distribution of micronutrient deficiencies and shifts in dietary patterns. Additionally, this study will provide information on emerging determinants such as mental health and intake of nutrients such as folate and $B_{12}$ that have not been included in previous studies. Finally, the inclusion of adolescent girls and school-age children will provide vital information on nutritional indicators for these target groups, which are often not included in other nationally representative surveys. This survey will also provide information on the coverage of direct interventions implemented in the health sectors and indirect interventions implemented in the agriculture, Water, Sanitation and Hygiene (WASH), education and social protection sectors for whom scant data exist. Hence, this study will provide valuable information that will guide the implementation of strategic actions for the reduction of malnutrition in Ethiopia.

**Contributors** MT, AL, SC, MW, AP, AS and MG conceived the study and drafted the original protocol. All authors participated in refining the protocol. AH, MW, MG and MT played a major role in statistical consideration. DAD, FC, MG, RN and ASD helped to write the draft protocol and made critical contribution to the content. KB, AL, GT, MHD, LT and MZ supervised manuscript preparation. All authors were responsible for reviewing, final editing and approval of the manuscript.

**Funding** This work was supported by Power of Nutrition,This work was supported by Power of Nutrition,

**Patient consent for publication** Consent obtained directly from patient(s)

**ORCID iDs**
Meron Girma http://orcid.org/0000-0001-9114-1412
Masresha Tessema http://orcid.org/0000-0002-7155-4815

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
