## [Reviewer comments · BMJ Open]

ARTICLE DETAILS

TITLE (PROVISIONAL)	Ethiopia National Food and Nutrition Survey to inform the Ethiopian National Food and Nutrition Strategy: a study protocol
AUTHORS	Woldeyohannes, Meseret; Girma, Meron; Petros, Alemnesh; Hussen, Alemayehu; Samuel, Aregash; Dinssa, Danial; Challa, Feyissa; Lailou, Arnaud; Chitekwe, Stanley; Baye, Kaleab; Noor, Ramadhani; Donze, Anne Sophie; Tollera, Getachew; Dangiso, Mesay; Tadesse, Lia; Zelalem, Meseret; Tessema, Masresha

VERSION 1 – REVIEW

REVIEWER	Nguyen , T FHI 360, Alive & Thrive
REVIEW RETURNED	25-Oct-2022

GENERAL COMMENTS	1. The authors summarized a complex nationally representative study that covers various aspects of nutrition and health. I have a few comments and suggestions:2. In the introduction of the abstract (lines 23-25), please state specific diseases and their risk factors.3. Line 41, please mention that it is a full review.4. Lines 46-49, in which context did the authors mentioned: Ethiopia, East Africa, or globally?5. Line 61, one in three (missing population)?6. Line 111, this objective is not clear. Nutrition-specific and nutrition-sensitive interventions are very broad. Corresponding indicators did not fit the “interventions.” I have the same comments in table 1 and the text.7. Sampling might be good to have a diagram.8. Methods will be better if the authors include Questionnaires and lab protocol as supplementary materials. What is the procedure for informed consent?9. What are the cut points for BMI for adults?10. I am not sure about BMJOpen. However, some journals asked for a summary of the progress of data collection to date.
--

REVIEWER	Otekunrin , Olutosin A. Federal University of Agriculture Abeokuta
REVIEW RETURNED	21-Jan-2023

GENERAL COMMENTS	This is an interesting study protocol. As regards the study participants, will the household heads be only female or can be male or female? Please specify. This is because the adult male members were not part of the categories for this study. The sample size description and estimation were not clear enough. Authors are expected to present a detail description of the sampling procedures for the study.
---

	From Table 2S, how did you arrive at the expected number of the categories of household members? Remember that the total population of the regions were not included in the table. Kindly add the population of the regions. What's the difference between under-five children and 6-59months children? In Table 2S, the category of 6-12 years old is missing while 6-59 months children were not part of the categories specified in the main text. Under the dietary assessment section, authors are expected to present the food groups for each of the household categories. Further, under anthropometric status, you may need to add the software that will be used to calculate the anthropometric indices for these groups especially the children and the adolescents. The under-five and 5-9 years old children using WHO Anthro while adolescents may use WHO Anthroplus. Thank you.
--	--

REVIEWER	Dassey, Vincent Nutrition Services Section Ministry of Health, Community Development, Gender, Elderly and Children
REVIEW RETURNED	30-Jan-2023

GENERAL COMMENTS	Excellent protocol preparation, with additional thought to look at the agricultural soil nutrients, will be the first time extensive surveys include a study on soil micronutrients and associate with the direct and indirect parameters of micronutrient intake. I wish you good luck in implementing this protocol.
--

VERSION 1 – AUTHOR RESPONSE

Reviewer 1 Comments	Our Responses (in the new submission)
The authors summarized a complex nationally representative study that covers various aspects of nutrition and health. I have a few comments and suggestions:	Thank you.
In the introduction of the abstract (lines 23-25), please state specific diseases and their risk factors.	We have now made edition and added specific diseases and their risk factors. The below text added in the revised version of Manuscript: 'Despite such improvements, a substantial segment of the country's population remains chronically undernourished and suffers from not only micronutrient deficiencies (Zinc, Iodine, Vitamin A and D, and Iron) but also from increasing diet-related non-communicable diseases such as diabetics, hypertension and cancer.'
Line 41, please mention that it is a full review.	Yes, the study was full reviewed by the Ethiopian Public Health Institute Institutional Review Board (IRB). We have added full review in the revised protocol.
Lines 46-49, in which context did the authors mentioned: Ethiopia, East Africa, or globally?	Indeed the context was written in Ethiopia. We have now more clarified it.
Line 61, one in three (missing population)?	Thank you for ratifying this error. We have now corrected the sentence as stated below: 'Globally,

	one in every three population are affected by one of more forms of malnutrition.'
Line 111, this objective is not clear. Nutrition-specific and nutrition-sensitive interventions are very broad. Corresponding indicators did not fit the "interventions." I have the same comments in table 1 and the text.	We have modified it as direct and nutrition indirect interventions. We have also added table as table 1 which indicates all coverage's with target group.
Sampling might be good to have a diagram.	We have drawn diagram that shows the sampling frame. It appears as figure 2 in the updated Manuscript.
Methods will be better if the authors include Questionnaires and lab protocol as supplementary materials. What is the procedure for informed consent?	We have added data collection tool as Supplementary text 2 , Laboratories protocols and SOP as Supplementary text 3-9. We have also added informed consent in the supplementary text 1
What is the procedure for informed consent?	Upon arrival to individual household, our field data collectors will get consent from each household head. Subsequently, they will also get from other member. The detail about the consent is attached in the supplementary text 1
What are the cut points for BMI for adults?	We will be using 18.5 which also WHO recommendation.
I am not sure about BMJOpen. However, some journals asked for a summary of the progress of data collection to date. [NOTE FROM THE EDITORS: While no specific detail is required, we do request that the status of the study and the planned timeline for its completion be indicated in the main text of protocol manuscripts]	During our submission of Manuscript, we were started data collection. Now, we have completed data collection in 11 regions of the total of 12 regions. This has been indicated in the submission form.
Reviewer 2 Comments	Our Responses (in the new submission)
This is an interesting study protocol. As regards the study participants, will the household heads be only female or can be male or female? Please specify. This is because the adult male members were not part of the categories for this study. The sample size description and estimation were not clear enough. Authors are expected to present a detail description of the sampling procedures for the study.	Household head will be male or female headed household You have raised an important point here. Household will be part of our study. We will be asking household head about household characteristic, and socio economic status. However, we will not collect biomarkers and dietary assessment from household head if they are male headed household head.
From Table 2S, how did you arrive at the expected number of the categories of household members? Remember that the total population of the regions were not included in the table. Kindly add the population of the regions.	We have added population in the revised version of manuscript.
What's the difference between under-five children and 6-59months children?	Apologies for the error. We have now corrected the table. Sorry for this confusion.
In Table 2S, the category of 6-12 years old is missing while 6-59 months children were not part of the categories specified in the main text.	Thank you for identifying the error. We have corrected the error the revised manuscript.
Under the dietary assessment section, authors are expected to present the food groups for each of the household categories.	We have added the following text from line 228-236. 'We will use 15 food groups to assess dietary intakes of women (15-49 months) and children

	aged 24-59 months. These food groups were: 1) Cereals and their products 2) Starchy Roots and tubers, and their products 3) Pulses, and their products 4) Vegetables and their products 5) Fruits and their products 6) Meat, and poultry their products 7) Eggs and their products 8) Fish, shellfish and their products 9) Milk and milk products 10) Fats and oils 11) Nuts and seeds 12) Sugar and sweetened products 13) Beverages 14) Spices and condiments and 15) Miscellaneous. For children aged 6-23 months we will use the updated WHO, UNICEF food groups: 1) Breastmilk, 2) Grains, roots and tubers 3) Pulses, nuts and seeds, 4) Dairy products 5) Flesh foods (meats, fish, poultry, organ meats) 6) Eggs 7) Vitamin-A rich fruits and vegetables; and 8) other fruits and vegetables.'
Further, under anthropometric status, you may need to add the software that will be used to calculate the anthropometric indices for these groups especially the children and the adolescents. The under-five and 5-9 years old children using WHO Anthro while adolescents may use WHO Anthroplus.	We added the following sentence in the analysis section in line number 338-339: "Anthropometric indices will be calculated using the WHO Anthro software for under five children and WHO AnthroPlus software for adolescent."
Reviewer 3 Comments	Our Responses (in the new submission)
Excellent protocol preparation, with additional thought to look at the agricultural soil nutrients, will be the first time extensive surveys include a study on soil micronutrients and associate with the direct and indirect parameters of micronutrient intake. I wish you good luck in implementing this protocol.	Thank you.

VERSION 2 – REVIEW

REVIEWER	Otegunrin , Olutosin A. Federal University of Agriculture Abeokuta
REVIEW RETURNED	24-Feb-2023

GENERAL COMMENTS	It is obvious that authors have made substantial changes to the original version of this manuscript. I have just one additional comment which centres on line 222-230. You have made it clear that the study will adopt a 15 food groups. I feel you need to add what informed this type of food groups. Do you have literature to support it. Are you adapting this from previous studies? Kindly add this information here. Thank you.
--

VERSION 2 – AUTHOR RESPONSE

Reviewer: 2 Dr. Olutosin A. Otegunrin , Federal University of Agriculture Abeokuta	The food groups were adapted from the FAO/WHO Global Individual Food consumption data Tool (GIFT) food
---	--

Comments to the Author: It is obvious that authors have made substantial changes to the original version of this manuscript. I have just one additional comment which centres on line 222-230. You have made it clear that the study will adopt a 15 food groups. I feel you need to add what informed this type of food groups. Do you have literature to support it. Are you adapting this from previous studies? Kindly add this information here. Thank you.	groups and food subgroup recommendations. Now, we have added this information to the manuscript.
Reviewer: 2 Competing interests of Reviewer: I declare that I do not have any competing interests.	Thank you!

VERSION 3 – REVIEW

REVIEWER	Otekunrin , Olutosin A. Federal University of Agriculture Abeokuta
REVIEW RETURNED	29-Mar-2023
GENERAL COMMENTS	Accept for publication.

VERSION 3 – AUTHOR RESPONSE